# Identifying Persons with Special Healthcare Needs in Dentistry—Development and Validation of the French Case Mix Tool

**DOI:** 10.3390/ijerph20042997

**Published:** 2023-02-08

**Authors:** Denise Faulks, Marie-Sophie Bogner, Solenn Hamon, Caroline Eschevins, Bruno Pereira

**Affiliations:** 1Centre de Recherche en Odontologie Clinique (CROC), Université Clermont Auvergne, 63100 Clermont-Ferrand, France; 2Service d’Odontologie, CHU Clermont-Ferrand, 63000 Clermont Ferrand, France; 3Dental Surgery, 2 Rue de la Poudrette, 69100 Villeurbanne, France; 4Direction de la Recherche Clinique et de l’Innovation, CHU Clermont-Ferrand, 63000 Clermont-Ferrand, France

**Keywords:** disabled persons, oral health, public health dentistry, health services research

## Abstract

Providing dental care for certain patient groups is complicated due to difficulties with cooperation, communication, health conditions, and social context, amongst others. The majority of dentists in France work within a public fee-per-item system. A new measure has been introduced providing a financial supplement to dentists for each episode of care for a patient with a severe disability. This supplement is justified by completion of the French Case Mix tool (FCM), a new measure designed to retrospectively identify episodes of dental care that have required adaptation and additional time or expertise. The aim of this study was to investigate the validity and psychometric properties of the FCM. The content validity of the tool was improved at each round of pilot development, involving 392 patient encounters. Test–retest data at 2 weeks for 12 fictional patient treatment episodes were collected from 51 dentists. This phase confirmed inter- and intra-dentist reproducibility, criterion validity, and interpretability. Retrospective analysis of 4814 treatment episodes nationally demonstrated high reliability, internal consistency, and construct validity. Overall, the FCM showed high validity and good psychometric properties. However, the impact of providing a financial supplement on improving access to care for persons with special needs has yet to be evaluated.

## 1. Introduction

Providing healthcare for certain patient groups is complicated due to difficulties with cooperation, communication, health conditions, and social context, amongst others [1]. Examples of groups affected include the dependent elderly, persons with disabilities, young infants, persons with mental health problems, incarcerated persons, homeless persons, etc. Such patient groups may be considered as having special needs in health care, as services must be adapted if high-quality, appropriate treatment is to be provided. Adaptation of services typically requires additional expertise, equipment, and time, and thus imposes additional costs on healthcare providers. In many health systems, dentistry is financed on a “fee per item” basis, that is, the dentist is paid a set fee by the state for a defined item of treatment. This payment system is considered to be incompatible with access to care for persons requiring additional time or expertise in dentistry [2]. To compensate for this problem, many public health systems provide salaried services for special care patients but this implies exclusion from the mainstream system, and the creation and financing of suitable and sufficient local salaried services. In addition, when commissioning services, it is difficult to identify and quantify needs when the target population is ill-defined and hard to reach [3,4].

In order to encourage the provision of care for populations with special healthcare needs, systems must be able to identify those persons requiring adapted care and must ensure appropriate financial compensation to the service provider for potential costs engendered. Although there is little consensus about the definition of “complexity of care” in the literature [5], it is clear that patient diagnosis alone does not adequately capture differences in care complexity [6]. In 2010, the British Dental Association proposed a special care case mix tool to address this problem [3]. Different healthcare systems and healthcare cultures require different tools, however, and the development of country-specific “case-mix” tools is growing (although these remain unpublished as yet).

In France, the vast majority of dentists (89%) work in general practice within the primary public healthcare system, on a fee-per-item basis [7]. There is no specific salaried service for special care dentistry. In order to encourage access to adapted dental care within the primary healthcare system, a financial supplement for dentists for each treatment session with a patient with a severe disability has recently been introduced. To justify the financial supplement, dentists are asked to complete the French Case Mix tool (FCM) (in French the FCM is entitled APECS—échelle des Adaptations pour une Prise En Charge Spécifique en odontologie). This tool was originally designed to identify the degree of adaptation necessary for appropriate dental treatment to be provided for all types of special care patients. The FCM had not been validated on its introduction into the dental fee structure.

The aim of the current study was to investigate the validity and psychometric properties of the French Case Mix tool in the identification of patients with special healthcare needs in dentistry. The specific objectives were to describe the development of the FCM; to describe pilot studies undertaken to improve content validity; to undertake a test–retest study to evaluate criterion validity, intra-dentist and inter-dentist reproducibility, and interpretability; and to retrospectively analyze national data from patient records to verify internal consistency, floor, and ceiling effects, and construct validity.

## 2. Materials and Methods

### 2.1. Development of the FCM

A group of experts from the French Association for Disability and Oral Health (SOSS Santé Orale et Soins Spécifiques, www.soss.fr, accessed on 30 January 2023) (termed “the national association” from here on) developed a preliminary version of the French Case Mix tool (FCM). A working group of three special care specialists, one in hospital practice, one in university hospital practice, and one in general practice, drew up an initial draft of the FCM following an informal scoping review of the relevant literature. This draft was debated, reworked, and reworded during a dedicated session of the Executive Board of the national association (made up of 12 persons including dental professionals, medico-social professionals, and patient representatives). This version underwent three waves of piloting by practitioners in two different specialist hospital practices and in one general practice. These pilot projects led to changes in wording and presentation. The draft FCM was presented to the Disability Working Group of the French Social Security Department working on the new financial measures for dental treatment (Caisse Nationale d’Assurance Maladie). Further amendments were made by the experts of this group including the addition of the American Society of Anesthesiologists classification for the evaluation of medical health [8].

The definitive version of the FCM is given in Appendix A. It consists of seven domains: Communication; need for sedation, general anesthesia (GA) or other facilitatory techniques; cooperation; medical health; dental risk factors; autonomy; administrative coordination. The FCM is designed to be completed by the dentist at the end of each treatment session. The dentist gives a score of no adaptation, minor adaptation, moderate adaptation, or major adaptation for each of the seven domains. If the treatment session involving a patient with a disability is scored as having required moderate or major adaptation in at least one domain, the dentist is eligible for a financial supplement.

### 2.2. Test–Retest Evaluation

A group of five experienced special care specialists from the national association with different practice backgrounds designed twelve virtual case studies, detailing treatment sessions with different types of patients and different levels of adaptation to standard dental care. Each case study was accompanied by a photograph of the fictional patient and a detailed description of the patient encounter, including all necessary information to complete the FCM. The experts completed the FCM for each case study individually and then reached a consensus through debate to create a set of “gold-standard” responses. The limits of the “gold- standard” are recognized in that, in practice, a dentist’s experience of the complexity of care is subjective and will vary according to his/her skill and experience in special care dentistry.

All members of the national association were solicited by email to take part in the test–retest study and to encourage their colleagues to participate in a snowballing process. All dentists responding to the call for participation were given a secure link to a website presenting the twelve virtual case studies. The dentists completed the FCM in relation to each case study. After 14 days, they were sent up to three email reminders to re-complete the FCM for the same case studies. The sample size was fixed according to COSMIN guidelines [9]. Participants were included until at least 50 test–retest data sets had been collected, in accordance with Terwee et al. [10] who suggest that a positive rating for reliability can be given when the ICC or weighted Kappa is at least 0.70 in a sample size of at least 50 participants.

### 2.3. National Data

All members of the national association were solicited to share data. FCM forms were retrieved from patient files of participating dentists, along with anonymized sociodemographic data (patient age, gender, patient with disability/dependent elderly/dental anxiety). Data collection took place for patients who had been treated between May 2019 and May 2020. FCM forms were included if they were fully completed. There were no exclusion criteria regarding patient characteristics (age, medical history, or social context for example). For each patient session, the dentist evaluated whether complex care had been required. Dentists described their type of practice (private/other), their target population (special care/general population), their affiliation to a special care coordination network, and their special care specialist status. The sample size was fixed according to COSMIN guidelines [9], and it was hoped that at least 50 dentists would provide up to 100 FCM forms each.

The protocol was certified by the Institutional Review Board of the University Hospital of Clermont Ferrand (Ref: 2019/CE38).

### 2.4. Statistical Analysis

The psychometric properties of the FCM were evaluated following COSMIN guidelines [9], using criteria described by Terwee and al. in 2007 [10]. Data from the test–retest and the national study were analyzed separately. Statistical analysis was performed using Stata13 (StataCorp, College Station, TX, USA). A summary of the analysis methodology is given in Table 1.

To compare socio-demographic characteristics, either the Chi^2^ or Fisher exact test was used.

Content validity was sought by analyzing the feedback given by dentists during the pilot. In particular, the time taken to complete the survey and the acceptability of the domain concepts were reported.

Internal consistency was assessed using Cronbach’s alpha. Item rest correlation, average inter-item correlation, and labeled alpha were calculated for each domain. The inter-item correlation matrix was also analyzed. It was important that all domains measured different concepts, so that how individual domains related to the overall scale was examined. A Cronbach’s alpha score between 0.7 and 0.8 was expected. The ideal range of average inter-item correlation should be 0.15 to 0.50.

Floor and ceiling effects were evaluated based on eligibility for the financial supplement (a score of moderate or major in at least one of the seven domains of the FCM). For each item, the percentage of patients needing minor and major adaptations was calculated.

Construct validity was evaluated by analyzing the agreement between the dentist’s opinion of care complexity and eligibility for the financial supplement using the weighted Kappa coefficient. In order to further investigate this concept, the Item Response Theory (IRT) was applied to estimate the properties of the domains and the utility of each domain. More precisely, the IRT Model was used to produce item information characteristic curves to evaluate whether the four-point Likert scale was relevant to the level of adaptation in each domain.

In terms of criterion validity, the correlation between the dentist’s scores and the “gold standard” replies in the test–retest study was measured, with a positive rating if the correlation was at least 0.70.

The test–retest study allowed for the investigation of reproducibility. The concordance coefficient (weighted Kappa) and concordance percentage were used to evaluate the intra-dentist agreement. A concordance coefficient less than 0.4 was considered low, above 0.7 good, and above 0.8 excellent. The inter-dentist agreement was calculated using the Intra-Class Correlation Coefficient (ICC). Close to 1, the ICC indicates a high degree of similarity between the responses of practitioners, whereas when it is close to 0, it shows responses that are less similar.

Interpretability was investigated by analyzing the dentist’s replies against the “gold standard” in the test–retest study in terms of whether the case study was considered eligible for the financial supplement or not.

To analyze the reliability of the domains to identify patients requiring adaptation, a receiver operating characteristic curve was constructed. The cut-off of at least one domain with a moderate or major score in order to be eligible for the financial supplement was an arbitrary decision. The area under the curve (AUC) was, therefore, calculated for the number of items with a moderate or major adaptation score in order to discuss this decision. AUC is an effective way to summarize the overall diagnostic accuracy of the test. The lower limit on the confidence interval is expected to be <0.7 [11]. The positive predictive value (PPV) and the negative predictive value (NPV) were examined to investigate whether the FCM agreed with the dentist’s appraisal of care complexity.

It was not appropriate to evaluate responsiveness in this study as the FCM is designed to be completed retrospectively after each treatment session, as a one-off event, not to measure change over time.

## 3. Results

### 3.1. Study Population

In the pilot studies, 392 draft FCM forms were completed in the three centers, with ongoing modification of the forms at each round.

In the test–retest study, 51 dentists completed the FCM for the 12 virtual case studies twice, at an interval of at least two weeks.

Nationally, 4814 FCM forms were included from 113 dentists in 21 different French départements (boroughs). Forms were included if the FCM data were complete, even if certain demographic data were missing. The percentage of forms relating to a female patient was 49% and the average age of the patients was 37.4 years ± 23.9. Forty-two percent of the forms were collected from dentists working in private practice and 44% were collected from dentists that only provided treatment for those with special needs. Thirty-five percent of the forms were collected from specialists in special care dentistry (Médicine Bucco-Dentaire). Over half (56%) of the forms were collected from dentists working within a care coordination network for patients with special needs. The distribution of different patient populations is given in Table 2 for the 4684 forms where this information was given.

Of the 4814 completed FCM forms, 3001 (62%) related to care that the dentist had considered “complex” and needed adaptation. Sixty percent (2903) were eligible for the financial supplement (a FCM score of moderate or major in at least one domain). The sociodemographics of patients needing adaptation on the FCM and those considered as “complex” were statistically similar, except for the distribution of patient type (*p* < 0.05).

Patients considered as “complex” and FCM forms with at least one score of moderate or major were significantly different from those with only minor or no adaptation, or not requiring “complex” care. Patients identified as requiring adapted care were on average younger, more likely to be male, more likely to be disabled, less likely to attend private practice, and more likely to see a specialist dentist or a dentist with a targeted patient population, or affiliated to a care coordination network (*p* < 0.001).

### 3.2. FCM Scores for the National Data by Domain

The distribution of moderate or major scores on the FCM by domain is given in Table 3 and in Figure 1. For those forms with at least one moderate or major score, the average number of domains scoring moderate or major was 1.8 ± 1.9 domains. In addition, scores were significantly higher in all domains when care had been considered “complex” (*p* < 0.001). Overall, the distribution of the scores was similar between those forms with at least one score of moderate or major and those forms for which the dentist had declared the care as “complex”.

### 3.3. Psychometric Analysis

#### 3.3.1. Content Validity

At the pilot stage, 96% of FCM forms were completed in under one minute.

Different waves of piloting confirmed that the concepts expressed by the seven domains were legitimate and sufficient. No new domains were suggested and no domains were identified as redundant. Changes in the wording were made gradually with feedback. The initial version of the FCM had weighted scores and was designed to give a cumulative total that would have been subject to cut-off in terms of eligibility for financial compensation. This structure was abandoned as the domains were designed to be independent and it was recognized that weighting the scores would penalize patients with a high score in one domain only. The weighted scoring system was replaced by a simple Likert scale. The domain regarding medical complexity proved the hardest to rate for the dentists, and formulation of this domain using the ASA scale was the last amendment [8]. By the end of piloting, no further suggestions for change were recorded.

#### 3.3.2. Internal Consistency

The results of the internal consistency analysis for the national data were very good, as expressed by a Cronbach’s alpha of 0.85. Cronbach’s alpha coefficients for each domain (labeled alpha) demonstrate how the alpha for the scale would change if the item were deleted from the scale. Labeled alpha results were high and similar (0.81 < alpha < 0.86), demonstrating that each different domain in the FCM was consistent with the others.

The item rest correlation rates demonstrate a correlation between an item and a scale made up of all the other items. The item rests correlation rates (Table 4) ranged from 0.45 (“medical health” domain) to 0.79 (“autonomy” domain) suggesting that the domains all contribute to the overall consistency of the scale. The average inter-item correlation rates ranged from 0.41 to 0.50. The inter-item correlation matrix gave values between 0.10 < r < 0.73. The average item-rest correlation results suggest that the domains are independent and measure different concepts. This is important as it implies the FCM can identify patients with high adaptation needs in one domain only, as well as those with needs in several domains. The domains “communication” and “autonomy” were the most correlated (r = 0.73), whereas the domains “need sedation/GA” and “medical health” were very poorly correlated (r = 0.10). These results suggest that all seven domains of the FCM should be retained.

#### 3.3.3. Floor and Ceiling Effects

Floor and ceiling effects were related to the proportion of forms eligible for the financial supplement. All domains had a floor effect above 15% (range 18–57%) and a small ceiling effect (range 3–14%) except for the “communication” (27%) and “coordination” domains (17%).

#### 3.3.4. Construct Validity

The FCM scores differed in relation to whether the treatment session had been considered by the dentist as “complex” or not (*p* < 0.001). In the national study, 89% of the sessions evaluated by dentists as “complex” were effectively identified by the FCM as being eligible for a financial supplement (at least one score of moderate or major) (Kappa 0.77). This suggests that the items of the FCM made sense to the dentists.

The IRT model was used to discriminate between the properties of each domain. All domains were considered discriminant and informative (*p* ≤ 0.001), particularly the domains “communication” (DIF = 4.42 [4.10; 4.73]) and “autonomy” (DIF = 4.55 [4.25; 4.87]). The “need for sedation/GA” (DIF = 1.54 [1.43; 1.64]) and “medical health” (DIF = 1.45 [1.35; 1.54]) domains provided less information at trait levels that remain above average (Figure 2).

Item information characteristic curves confirmed continuous ordering of thresholds of the four-point Likert scale. This suggested that all response categories are independent and make sense, thus despite the dichotomous outcome of the FCM (eligible/not eligible), the four-level score was considered appropriate (no adaptation, minor, moderate, and major adaptation) (Figure 3).

#### 3.3.5. Criterion Validity, Reproducibility and Interpretability

The results of the test–retest study are summarized in Table 5.

Correlation between the dentists’ answers and the “gold standard” answers ranged from 0.65 to 0.83 according to the domain, indicating a high level of criterion validity.

Intra-dentist reproducibility was expressed by a weighted kappa of 0.77 and a concordance coefficient of 80%, showing good concordance between the test and retest measures in all domains and all case studies combined. When the domains were analyzed separately, the kappa varied between 0.69 and 0.93. The inter-dentist agreement for each domain was high with an ICC of 0.70 [0.51–0.84] to 0.89 [0.79–0.95]. Those domains with greater variability may be considered as being the least easy to apply for dentists. Comparing the different domains for each 12 virtual case studies, concordance rates ranged from 61% to 100% for “communication”, from 80% to 100% for “cooperation”, from 63 to 100% for “need sedation/GA”, from 53 to 100% for “medical health”, from 53 to 100% for “dental risk”, from 69 to 100% for “autonomy”, and from 57% to 100% for “coordination”.

In terms of interpretability (Table 5), the virtual case studies were scored by the dentists in a similar way to the gold standard in terms of eligibility for the financial supplement. The percentage of domains scoring moderate or major ranged from 16% to 37% for dentists and from 8% to 42% for the “gold standard”. The difference in scores was ≤5%, except for the health status domain where the difference was 15%. The ASA scale used in this domain was perhaps insufficiently familiar to the dentists.

#### 3.3.6. Reliability

National data were used to analyze reliability. The area under the curve (AUC) showed very good reliability (0.92 [0.91; 0.93]) (Figure 4). This suggests a 92% chance that the dentist will correctly distinguish whether a patient requires adapted care or not, using the FCM. A cut-off point of at least one domain with a moderate or major score was shown to be the most appropriate for discriminating between levels of adaptation, with a sensitivity of 89% [88%; 90%] and specificity of 89% [87%; 90%]. The level of adaptation evaluated by the FCM is equivalent to the complexity of care as judged by the dentist (PPV 93% [92%; 94%]; NPV 83% [81%; 84%]).

## 4. Discussion

French policy makers introduced a compensatory financial measure for dental practitioners treating patients with “severe disability”. The French Case Mix Tool is written into the public fee scale and appears in all French dental payment software as a measure to justify this financial compensation but it had not yet been evaluated on introduction. This study describes the development and validation of the FCM through three phases: pilot studies, a test–retest study, and national data collection. The FCM tool was found to be valid and to have good psychometric properties. It was able to identify patients requiring adaptation for their dental care with very good reliability. Internal consistency was high and the seven domains demonstrated consistency with each other while remaining independent constructs. The agreement between the dentist’s opinion of the complexity of care and the FCM was very high, and the four-point Likert scale was found to be appropriate. The test–retest data gave evidence of inter-dentist and intra-dentist reproducibility and criterion validity. It was thus confirmed that the tool was fit for the purpose of retrospectively identifying dental treatment episodes where significant adaptation to standard care had been required.

The longest-standing measure of the complexity of patient care is the British Dental Association Case Mix tool (BDA) [3,12]. This tool is largely used to commission salaried public special care dental services in the UK [13]. No other published examples of case mix tools were found in the literature although the authors have knowledge of projects in both Australia and Malaysia, and the International Association of Disability and Oral Health (iADH) is planning to produce an international case mix tool. The nearest type of measure in the literature is represented by sporadic attempts to anticipate difficulty for certain patient groups, such as a tool to predict cooperation for dental patients with autism [14], those with Alzheimer’s disease [15]; or children with a disability [16]. Other tools have been used to estimate specific patient treatment needs and are used for commissioning care, for example, the IOTN for orthodontic services [17] or the IOSN for sedation services [4,18].

As far as the authors are aware, none of the above tools have been used to justify financial compensation at the individual patient level, as is the case in France. The original BDA case mix consisted of six domains with weighted ratings. It was designed to measure the relative complexity of care between different patients or of the same patient at different points in time. The outcome of the FCM, on the other hand, is measured in dichotomous terms: a score of moderate/severe in any one domain is sufficient to trigger justification of financial compensation, related to a specific treatment episode. As the domains have been shown to be independent, this system would seem fairer than using a weighted score in this context. Following the initial publication of the BDA tool, a simplified version was proposed, consisting of five domains [13,19]. It is possible that with ongoing evaluation of the FCM, it may be possible to simplify the tool in a similar manner in the future, although the completion time for the current tool was already reported at under one minute. It should also be underscored that the reply to the simple question of “Do you think this patient requires complex care or special care dentistry?” was highly correlated with the result of the FCM tool. It might, therefore, be appropriate to do away with the FCM altogether and consider that the dentist is competent to judge the level of complexity of care without the aid of a scale in the future. The FCM was not originally designed as a tool to ration access to financial compensation, but rather as a tool to quantify and qualify the adaptation to care necessary in special care dentistry. In this context, it was to be used primarily for research, epidemiology, or teaching. This debate is particularly pertinent in France, where medical practitioners are free to bill complex consultations at a higher financial fee without justification, but dental practitioners are asked to complete the FCM to justify the complexity.

Within the French dental healthcare system, the recognition of the additional expertise and time required to treat patients with special needs was welcomed as a major breakthrough. In a system dominated by general practitioners working in individual practices, it was hoped that the provision of a financial supplement, over and above the fee-per-item payment, would give incentive to dentists to treat patients from these populations. The FCM was originally designed to identify all situations where adaptation is necessary, including patients with severe dental anxiety, young children, homeless persons, persons with mental health problems, and the dependent elderly, for example, although the national study reported here did not explicitly describe all these groups. Unfortunately, the financial measure is currently limited solely to patients with “severe disability”, as this population was the explicit target of the new measure. The national association hopes that the financial measure will be extended to all patient groups requiring adapted care in the future in order to cover the full scope of special care dentistry. No figures are yet available to describe the demand for the financial supplement amongst French dentists, but anecdotal evidence is that uptake has been very poor. This is partly due to the lack of visibility of the new measure, which was introduced as a very small feature of the entire overhaul of the dental fee scale. In addition, its introduction coincided with the COVID-19 pandemic and was little discussed, even in the specialist media. For these reasons, further time and analysis are needed to evaluate the real public health impact of providing financial compensation for the additional expertise and time required to provide care for certain patient groups.

This study has certain strengths and limitations. The FCM is the first of such tools to undergo a stringent, structured investigation into its validity and psychometric properties. Unfortunately, this evaluation came after the measure had already been introduced into French law which gives a rather upside-down example of policymaking. Although every effort was made by the national association to ensure wide consultation in the development of the tool, the last word was given to the French Social Security Department which imposed certain amendments. The test–retest evaluation depended on the notion of a “gold standard”, which is arbitrary in the case of an individual practitioner’s perception of the complexity of care, as stated in the Method section. The analysis of the national data is strengthened by the large sample size. However, data were collected from members of the national association who were therefore aware of the complexities of special care dentistry. The results may have been different if data had been retrieved from general practitioners with no special interest in these patient populations. Finally, as mentioned above, the demographic categories available to characterize patients in the national study were unnecessarily restrictive.

## 5. Conclusions

The French Case Mix tool was found to show high validity and good psychometric properties in retrospectively identifying episodes of dental treatment where significant adaptation to standard care had been required. Despite the validity of the tool, further research is needed to assess the impact of the new dental public health measure of financial compensation in terms of improving access to dental care for persons requiring special care dentistry.

## Figures and Tables

**Figure 1 ijerph-20-02997-f001:**
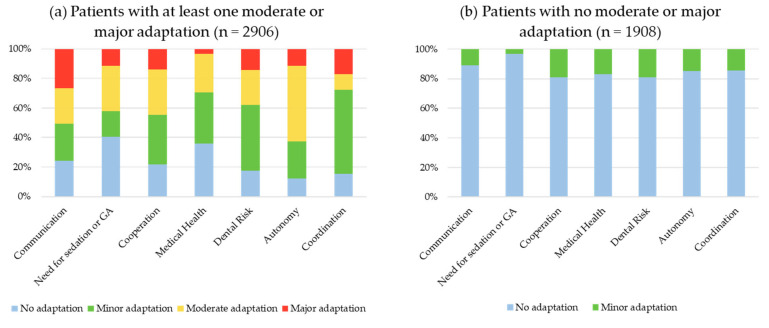
Figure 1 shows the distribution of scores: (**a**) distribution of scores for FCM forms with at least one moderate or major adaptation scored; (**b**) distribution of scores for FCM forms with no moderate or major adaptation scored.

**Figure 2 ijerph-20-02997-f002:**
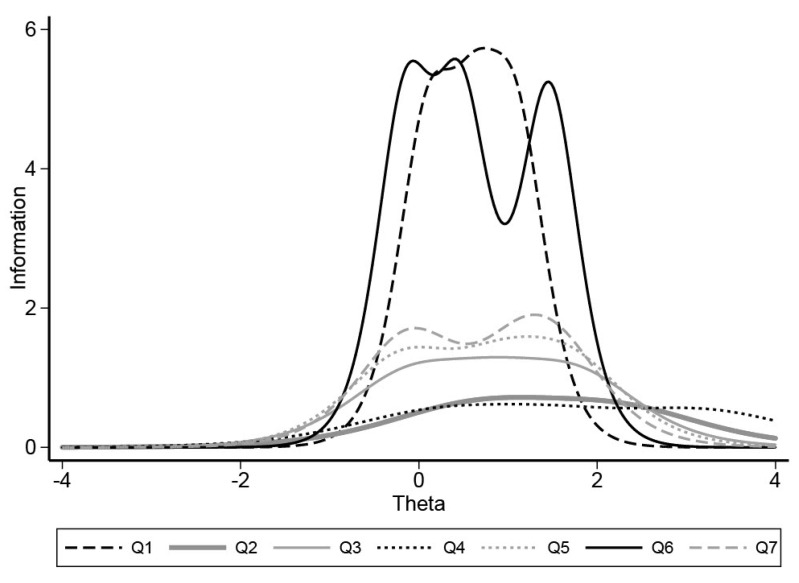
Figure 2 shows the item information functions for the seven domains of the FCM.

**Figure 3 ijerph-20-02997-f003:**
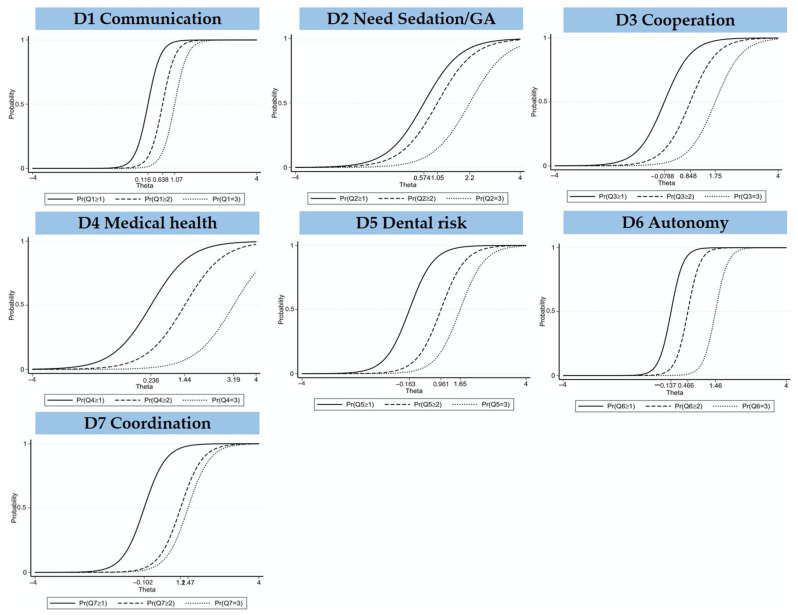
Item characteristic curves of the seven domains of the FCM.

**Figure 4 ijerph-20-02997-f004:**
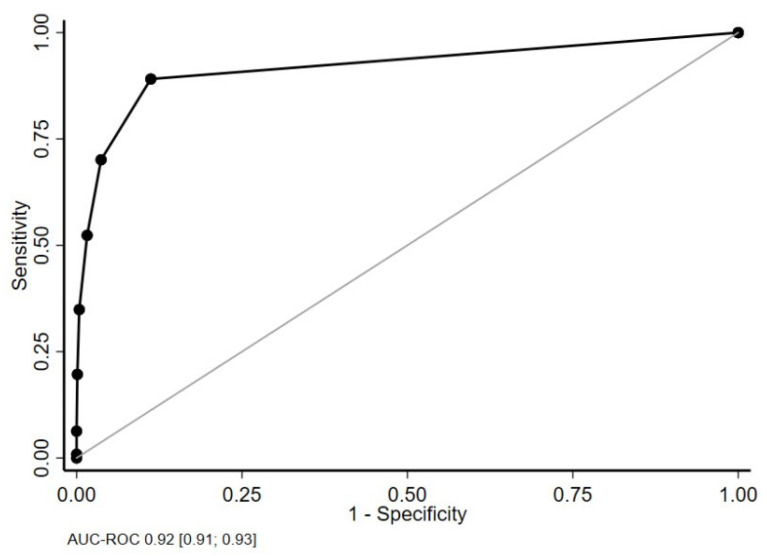
Receiver operating characteristic curve for the national data.

**Table 1 ijerph-20-02997-t001:** Summary of the analysis undertaken to validate the French Case Mix (FCM).

	Number of FCM Forms	Number of Dentists	Content Validity	Internal Consistency	Floor and Ceiling Effects	Construct Validity	Criterion Validity	Reproducibility (Inter-Dentist)	Reproducibility (Intra-Dentist)	Interpretability	Reliability
Pilot studies	392	unknown	x								
Test–retest study	12 × 2	51					x	x	x	x	
National data	4814	113		x	x	x					x

**Table 2 ijerph-20-02997-t002:** Distribution of FCM forms relating to different patient populations.

	N° of FCM Forms (% of Total)	N° of FCM Forms with Moderate or Major Adaptation (% of Population)
General population	1426 (30%)	84 (3%)
Patients with disability	2575 (55%)	2294 (79%)
Patients with dental anxiety	479 (10%)	346 (12%)
Dependent elderly patients	204 (4%)	164 (6%)
**TOTAL**	**4684 (100%)**	**2887 (100%)**

**Table 3 ijerph-20-02997-t003:** Distribution of moderate or major FCM scores.

FCM Domain	N° of Moderate or Major Scores (n = 4814)	% of Moderate or Major Scores
Communication	1476	31%
Need for sedation or GA	1224	25%
Cooperation	1300	27%
Medical health	857	18%
Dental risk	1100	23%
Autonomy	1820	38%
Coordination	806	17%

**Table 4 ijerph-20-02997-t004:** Summary of the internal consistency analysis for the national study.

Domains	Inter-Item Correlation (r)							Item-Rest Correlation	Average Inter-Item Correlation
	D1	D2	D3	D4	D5	D6	D7		
**D1 Communication**	1.00							0.76	0.42
**D2 Need Sedation/GA**	0.43	1.00						0.47	0.50
**D3 Cooperation**	0.63	0.59	1.00					0.64	0.45
**D4 Medical health**	0.40	0.10	0.25	1.00				0.45	0.50
**D5 Dental risk**	0.55	0.35	0.44	0.48	1.00			0.65	0.45
**D6 Autonomy**	0.73	0.38	0.54	0.49	0.61	1.00		0.79	0.41
**D7 Coordination**	0.52	0.28	0.38	0.36	0.45	0.63	1.00	0.59	0.46
**Test scale based on all items**									0.46

**Table 5 ijerph-20-02997-t005:** Summary of the test–retest analysis.

Domain	Criterion Validity	Reproducibility		Interpretability	
	Correlation with Gold Standard (Kappa) (% Concordance)	Inter-Dentist Concordance (ICC)	Intra-Dentist Correlation between Test and Retest (Kappa) (% Concordance)	% Case Studies Eligible for Supplement by Dentists	% Case Studies Eligible for Supplement by Gold Standard
**D1 Communication**	0.75 (85%)	0.87 [0.75–0.94]	0.78 (87%)	22%	25%
**D2 Need Sedation/GA ***	0.65 (78%)	0.89 [0.79–0.95]	0.93 (96%)	35%	33%
**D3 Cooperation**	0.83 (91%)	0.69 [0.82–0.96]	0.80 (89%)	18%	17%
**D4 Medical health**	0.67 (79%)	0.73 [0.54–0.86]	0.70 (80%)	23%	8%
**D5 Dental risk**	0.69 (77%)	0.75 [0.57–0.87]	0.74 (81%)	37%	42%
**D6 Autonomy**	0.67 (78%)	0.76 [0.59–0.88]	0.72 (82%)	22%	17%
**D7 Coordination**	0.67 (79%)	0.70 [0.51–0.84]	0.69 (81%)	16%	17%

* GA—General anaesthesia.

## Data Availability

The data presented in this study are available on request from the corresponding author. The data are not publicly available as this is not part of the French data protection agency protocol (CNIL) (registration number AN201102).

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
