# Peer review of "Identifying Persons with Special Healthcare Needs in Dentistry—Development and Validation of the French Case Mix Tool"

_ijerph, 2023, doi:10.3390/ijerph20042997_

Round 1

Reviewer 1 Report

Summary: 

This manuscript describes the validation of the French Case Mix (FCM) Tool to determine case complexity associated with the need for adaptations to care. The domains of the tool include communication, the need for sedation, cooperation, medical health, dental risk, autonomy, and coordination. The rationale for this study is to validate a tool whereby providers can receive enhanced funding for the care of patients with severe disability, specifically. However, the tool is designed to address multiple populations requiring adaptations (not limited to severe disability). Examples given of these populations include dependent elderly, persons with disability, young infants, persons with mental health problems, incarcerated persons, and homeless populations. The FCM Tool was developed by experts in the field of Special Care Dentistry. The tool was piloted and underwent further testing for validation following the COSMIN guidelines. The FCM Tool was shown to have high validity. 

General comments:

Overall this is a high-quality manuscript that adds to the field. The article is well-written and well-structured with excellent figures and tables. The validation methods are thorough and well-described, using the COSMIN guidelines. The rationale for validating the FCM Tool is clear. The tool is linked to reimbursement and there is only one other case mix tool in the literature, specific to the UK. The potential impact is high given the challenges with access to care among patients with complex needs. The disability language and concepts are thoughtful, recognizing the diversity of patients with disability. 

There are a couple areas that should be addressed: 

1. The development of the tool is not fully described. It is clear who developed the tool, but not how the initial version of the tool was developed. This could be reconciled by changing the title to address only validation and not development of the tool. Alternatively, a description of the process for development of the tool could be included in the manuscript. 

2. A description of limitations would be helpful. It appears the tool was meant to be applied to any populations that might need adaptations to care in the domains given. However, the national data seems to be limited to whether or not the patient fell into the general population, a patient with a disability, a patient with dental anxiety, and dependent elderly patients. It is unclear to what extent individuals experiencing homelessness, or individuals with mental health problems, or other relevant populations were included. 

Specific comments:

1. Line 52-53: This is an important statement/concept and needs to be retained in the final manuscript (…it is clear that patient diagnosis alone does not adequately capture differences in care complexity…”)

2. Line 59-64: The manuscript makes a strong case for the case tool being applicable to multiple populations. In this section the manuscript describes the use of the tool specifically with patients with severe disability. It would help to make it more clear in this section despite the tool being currently limited to patients with severe disability, the tool is designed to identify all situations where adaptations are necessary. This was more clear in lines 359-362. 

3. Table 1: This is an excellent and clear overview of the validation process throughout the various phases of the study. The table enhances readability of the text. Please retain this table in final draft. 

4. Line 349-353: Expand on conclusion here and the nuances of provider labeling of complexity vs use a validated tool in a research study vs when being reimbursed for care. 

Author Response

The authors would like to thank both reviewers for their constructive comments, detailed reviews and kind words.

Reviewer 1

General comments:

  1. The development of the tool is not fully described. It is clear who developed the tool, but not how the initial version of the tool was developed. This could be reconciled by changing the title to address only validation and not development of the tool. Alternatively, a description of the process for development of the tool could be included in the manuscript.

The following sentences have been added to clarify the development process:

“A working group of three special care specialists, one in hospital practice, one in university hospital practice and one in general practice, drew up an initial draft of the FCM following an informal scoping review of the relevant literature. This draft was debated, reworked and reworded during a dedicated session of the Executive board of the national association (made up of 12 persons including dental professionals, medico-social professionals and patient representatives).”

  1. A description of limitations would be helpful. It appears the tool was meant to be applied to any populations that might need adaptations to care in the domains given. However, the national data seems to be limited to whether or not the patient fell into the general population, a patient with a disability, a patient with dental anxiety, and dependent elderly patients. It is unclear to what extent individuals experiencing homelessness, or individuals with mental health problems, or other relevant populations were included.

The FCM was purposefully designed to capture all populations requiring adaptation to ‘standard’ care for any reason, as described. Unfortunately, the national study only collected patient characteristics under the headings of general population / patient with disability / dependent elderly / dental anxiety (Table 2). In retrospect, it would have been helpful to have an ‘Other’ category with a free field to capture more detail of other relevant populations. Details of the inclusion criteria have been added to the manuscript:

“FCM forms were included if they were fully completed. There were no exclusion criteria regarding patient characteristics (age, medical history or social context for example).”

This limitation has been added to the fourth paragraph of the discussion:

“Within the French dental healthcare system, the recognition of the additional expertise and time required to treat patients with special needs was welcomed as a major breakthrough. In a system dominated by general practitioners working in individual practices, it was hoped that the provision of a financial supplement, over and above the fee-per-item payment, would give incentive to dentists to treat patients from these populations. The FCM was originally designed to identify all situations where adaptation is necessary, including patients with severe dental anxiety, young children, homeless persons, persons with mental health problems, and the dependent elderly for example, although the national study reported here did not explicitly describe all these groups. Unfortunately, the financial measure is currently limited solely to patients with ‘severe disability’, as this population was the explicit target of the new measure. The national special care association hopes that the financial measure will be extended to all patient groups requiring adapted care in the future in order to cover the full scope of special care dentistry.”

Specific comments:

  1. Line 52-53: This is an important statement/concept and needs to be retained in the final manuscript

The authors agree.

  1. Line 59-64: The manuscript makes a strong case for the case tool being applicable to multiple populations. In this section the manuscript describes the use of the tool specifically with patients with severe disability. It would help to make it more clear in this section despite the tool being currently limited to patients with severe disability, the tool is designed to identify all situations where adaptations are necessary.

The manuscript has been adjusted to read:

               “This tool was originally designed to identify the degree of adaptation necessary for appropriate dental treatment to be provided for all types of special care patients.”

  1. Table 1: Please retain this table in final draft.

The authors agree.

  1. Line 349-353: Expand on conclusion here and the nuances of provider labeling of complexity vs use a validated tool in a research study vs when being reimbursed for care.

This concept has been expanded on as follows:

“The FCM was not originally designed as a tool to ration access to financial compensation, but rather as a tool to quantify and qualify the adaptation to care necessary in Special Care dentistry. In this context it was to be used primarily for research, epidemiology or teaching. This debate is particularly pertinent in France, where medical practitioners are free to bill complex consultations at a higher financial fee without justification, but dental practitioners are asked to complete the FCM to justify complexity.”

Authors’ corrections

The authors identified two typo errors: both in the Abstract and in Table 1, the number of treatment episodes analysed has been corrected to 4814 (and not 1484 erroneously stated).

Reviewer 2 Report

The research idea appears to be novel, however, some edits are required in the manuscript to make it comprehensive for its readers.

Abstract (Page 1)

Page 1: A structured abstract with relevant subheadings can be given (the journal guidelines must be looked into).  A general description of relevant study sample characteristics (geographic location, language, other relevant demographic and baseline characteristics) is missing.

Introduction

Page 2 Line 54-56: Reference needs to be added for this statement. A detailed literature review or any recent review of all the existing evidence regarding the care mix tools / French Case Mix(FCM) tool, in particular, can be added.  

Page 2 Line 62: Details regarding the FCM Tool like the different domains, number of items, scoring algorithm, the time period to complete the tool, and number or type of response categories can be added. 

Page 2 Line 67-69: The aim of the study looks fine but the specific objective(s) of the research needs to be mentioned in detail. 

Materials and methods

Page 2 Line 75: the characteristics of the experts' inclusion/ selection can be mentioned (if applicable). 

Page 3 Line 95: Details of the 5 experts (Branch/ specialization/ selection criteria) can be added.   

Page 3 Line 100: How the consensus was reached among the experts is missing. 

Page 3 Line 109: How the sample size of at least 50 test-retest data sets was arrived at is not mentioned. An explanation needs to be added for this statement.    

Sampling: As this study contains analyses using different sample sizes for different measures, the sample size and sampling technique for each analysis need to be mentioned in detail.  

Results

Page 5 Line 186: Sentence is starting with a number (42% of.....), hence needs to be re-framed. 

Page 7 Line 247-49: According to Table 4, the domains D2 Need Sedation/GA and D4 Medical Health were very poorly correlated (r=0.10). There is a mismatch between the text and results in the table.

Page 8 Line 282: There is a mismatch between the text and results in table 5. The correlation between the dentists’ answers and the ‘gold standard’ answers ranged from 0.65 (not 0.67) to 0.83 ...........

Discussion

Page 10 Line 314-24: The first paragraph can be reframed to include a summary of the rationale of the study, what was done, and then a brief summary of the major findings. Currently, it is a duplicate of only the results section. 

Page 11 lines 359-72: Strengths and limitations specific to the study should be discussed more in detail as the limitations mentioned are more generic in nature.

Author Response

The authors would like to thank both reviewers for their constructive comments, detailed reviews and kind words.

Reviewer 2

General comments

  1. Page 1: A structured abstract with relevant subheadings can be given (the journal guidelines must be looked into). A general description of relevant study sample characteristics (geographic location, language, other relevant demographic and baseline characteristics) is missing.

The instructions to authors state that the abstract “should follow the style of structured abstracts, but without headings”. As such, no subheadings have been added. The authors feel that the geographic location and language are clear (it is clearly a French national study) and that it would be difficult to go further into the demographics of the different study samples in 200 words. If the editor feels this is necessary, however, please let us know.

  1. Page 2 Line 54-56: Reference needs to be added for this statement. A detailed literature review or any recent review of all the existing evidence regarding the care mix tools / French Case Mix(FCM) tool, in particular, can be added.

This point is addressed in full in the 2nd paragraph of the discussion. There is no published review of existing Case Mix tools. The authors have undertaken such a review but it is in preparation for submission. Apart from the BDA tool cited in the text, no other descriptions of tools have been published in the scientific journals. A seminar at the 2022 International Association of Disability and oral Health conference addressed this subject.

  1. Page 2 Line 62: Details regarding the FCM Tool like the different domains, number of items, scoring algorithm, the time period to complete the tool, and number or type of response categories can be added.

This information is given in the 2nd paragraph of the Materials section.

  1. Page 2 Line 67-69: The aim of the study looks fine but the specific objective(s) of the research needs to be mentioned in detail.

The following details have been added:

“The specific objectives were to describe the development of the FCM; to undertake pilot studies to improve content validity; to undertake a test-retest study to evaluate criterion validity, intra-dentist and inter-dentist reproducibility, and interpretability; and to retrospectively analyse national data to verify internal consistency, floor and ceiling effects and construct validity.”

               With respect to this change and in order to avoid repetition, the first line of the Methods section has been removed

  1. Page 2 Line 75: the characteristics of the experts' inclusion/ selection can be mentioned (if applicable).

The following sentences have been added to clarify the development process:

“A working group of three special care specialists, one in hospital practice, one in university hospital practice and one in general practice, drew up an initial draft of the FCM following an informal scoping review of the relevant literature. This draft was debated, reworked and reworded during a dedicated session of the Executive board of the national association (made up of 12 persons including dental professionals, medico-social professionals and patient representatives).”

  1. Page 3 Line 95: Details of the 5 experts (Branch/ specialization/ selection criteria) can be added.

The sentence has been altered to read:

“A group of five experienced Special Care specialists from the national association with different practice backgrounds designed twelve virtual case studies, …”

  1. Page 3 Line 100: How the consensus was reached among the experts is missing.

This sentence has been altered as follows:

“The experts completed the FCM for each case study individually and then reached consensus through debate to create a set of ‘gold-standard’ responses.”

  1. Page 3 Line 109: How the sample size of at least 50 test-retest data sets was arrived at is not mentioned. An explanation needs to be added for this statement.

Sample size was fixed according to COSMIN guidelines (https://www.cosmin.nl/). More precisely, for test-retest analysis (reproducibility), the following sentence has been completed:

“Participants were included until at least 50 test-retest data sets had been collected, in accordance with Terwee et al. [10] who suggest that a positive rating for reliability can be given when the ICC or weighted Kappa is at least 0.70 in a sample size of at least 50 participants.”

  1. Sampling: As this study contains analyses using different sample sizes for different measures, the sample size and sampling technique for each analysis need to be mentioned in detail.

As aforementioned, the sample size was fixed according to COSMIN guidelines (https://www.cosmin.nl/). The following sentence has been added:

“The sample size was fixed according to COSMIN guidelines [9] and it was hoped that at least 50 dentists would provide up to 100 FCM forms each.”

  1. Page 5 Line 186: Sentence is starting with a number (42% of.....), hence needs to be re-framed.

This correction has been made.

  1. Page 7 Line 247-49: According to Table 4, the domains D2 Need Sedation/GA and D4 Medical Health were very poorly correlated (r=0.10). There is a mismatch between the text and results in the table.

The authors would like to thank the reviewer for finding this mistake, which has been corrected in the text, “Cooperation” being replaced by “Need Sedation/GA”.

  1. Page 8 Line 282: There is a mismatch between the text and results in table 5. The correlation between the dentists’ answers and the ‘gold standard’ answers ranged from 0.65 (not 0.67) to 0.83

Again, thankyou to the reviewer for pointing this out – the sentence has been corrected.

  1. Page 10 Line 314-24: The first paragraph can be reframed to include a summary of the rationale of the study, what was done, and then a brief summary of the major findings. Currently, it is a duplicate of only the results section.

The first paragraph of the discussion has been completed as follows:

“French policy makers introduced a compensatory financial measure for dental practitioners treating patients with ‘severe disability’. The French Case Mix Tool is written into the public fee scale and appears in all French dental payment software as a measure to justify this financial compensation but it had not yet been evaluated. This study describes the development and evaluation of the FCM through three phases: pilot studies, a test-retest study, and national data collection.”

  1. Page 11 lines 359-72: Strengths and limitations specific to the study should be discussed more in detail as the limitations mentioned are more generic in nature.

A paragraph has been added to the discussion:

“This study has certain strengths and limitations. The FCM is the first of such tools to undergo stringent, structured investigation into its validity and psychometric properties. Unfortunately, this evaluation came after the measure had already been introduced into French law which gives a rather upside-down example of policy making. Although every effort was made by the national association to ensure wide consultation in the development of the tool, the last word was given to the French social security department which made certain amendments without allowing time for re-piloting. The test-retest evaluation depended on the notion of a ‘gold-standard’, which is arbitrary in the case of an individual practitioner’s perception of complexity of care, as stated in the Method. The analysis of the national data is strengthened by the large sample size. However, data was collected from members of the national association who were therefore aware of the complexities of Special Care dentistry. The results may have been different if data had been retrieved from general practitioners with no special interest in these patient populations. Finally, as mentioned above, the demographic categories available to characterise patients in the national study were unnecessarily restrictive.”

Authors’ corrections

The authors identified two typo errors: both in the Abstract and in Table 1, the number of treatment episodes analysed has been corrected to 4814 (and not 1484 erroneously stated).